# Negative Emotions Will Be Welcomed: The Effect of Upward Comparison on Counterhedonic Consumption

**DOI:** 10.3390/bs14050374

**Published:** 2024-04-29

**Authors:** Shichang Liang, Tingting Zhang, Jingyi Li, Yiwei Zhang, Yu Tang, Lehua Bi, Feng Hu, Xueying Yuan

**Affiliations:** 1School of Business, Guangxi University, Nanning 530004, China; liangshch@gxu.edu.cn (S.L.); 2229301051@st.gxu.edu.cn (T.Z.); zhangyw0607@st.gxu.edu.cn (Y.Z.); 20160222@gxu.edu.cn (Y.T.); grzhf70@gxu.edu.cn (F.H.); yxy1118@st.gxu.edu.cn (X.Y.); 2School of Economics, Guangxi University, Nanning 530004, China; billsys@gxu.edu.cn

**Keywords:** upward comparison, counterhedonic consumption, relative deprivation, comparison target, horror consumption

## Abstract

Upward comparisons are prevalent in life and have a significant influence on consumer psychology and subsequent behavior. Previous research examined the effects of upward comparisons on consumption behavior, mainly focusing on behavior that evokes positive emotions (e.g., donation behavior, sustainable consumption) or behavior that evokes negative emotions (e.g., impulsive consumption, compulsive consumption) and less on behavior that evokes both negative emotions and positive emotions (i.e., counterhedonic consumption). This research examined the effect of upward comparisons on counterhedonic consumption. Five studies (N = 1111) demonstrated that upward comparison (vs. non-upward comparison) leads to counterhedonic consumption, and this effect is mediated by relative deprivation (Studies 2 and 3). In addition, this research showed that the comparison targets moderate the effects of upward comparisons on counterhedonic consumption. Specifically, when the comparison target is a friend, an upward comparison (vs. non-upward comparison) leads to counterhedonic consumption. When the comparison target is a stranger, an upward comparison (vs. non-upward comparison) has no significant influence on counterhedonic consumption (Study 5). Our findings extend the research on upward comparisons, relative deprivation, and counterhedonic consumption.

## 1. Introduction

From movies like *Spellbound 2* to music like *Dark Descent*, from horror games like *Quarry* to horror theme parks like Knott’s Berry Farm, people are attracted to purchasing and experiencing a wide range of products and services related to both negative and positive emotions. This type of consumption behavior is termed counterhedonic consumption, which describes the consumption of products or services that evoke both positive and negative emotions [1]. For example, engaging in scary activities on Halloween night can simultaneously evoke positive emotions such as happiness while instilling fear in consumers. Horror-themed escape rooms, likewise, can induce a range of emotions including nervousness, fear, excitement, and pleasure. In recent years, the market demand for this counterhedonic consumption has surged and received widespread attention. For instance, the National Retail Federation’s annual Halloween Consumer Survey reports that Halloween spending is expected to total a record USD 10.6 billion in 2022, surpassing the previous year’s record of USD 10.1 billion [2]. In China, horror-themed escape rooms are also popular. As horror-themed escape rooms continue to be updated and iterated, the immersive experience will attract more new consumers, and the market size is expected to reach CNY 17.59 billion in 2026 [3]. Upward comparisons are prevalent in life and have a significant influence on consumer psychology and subsequent behavior [4]. This research seeks to add to this understanding by exploring how upward comparisons may affect counterhedonic consumption.

Upward comparison occurs when people compare themselves to someone that they perceive to be superior [5]. Individuals have been frequently exposed to upward comparison events [4]. For example, employees may observe that colleagues receive higher salaries than themselves, and students may discover that classmates achieve higher grades than themselves. Scholars have examined the effects of upward comparisons on consumption behavior that evokes either positive or negative emotions. In research on consumption behaviors that can evoke negative emotions, scholars found that upward comparisons can lead to impulse consumption that can arouse negative emotions such as anxiety and regret [6,7], compulsive consumption that can arouse anxiety and regret [8,9], and indulgent food consumption that can awaken anxiety and guilt [10,11]. In research on consumption behaviors that can evoke positive emotions, scholars examined the effect of upward comparison on donation behavior that can improve self-esteem and increase well-being [12,13] and sustainable consumption that can make consumers feel pleasure [14,15]. Previous research examined the effects of upward comparisons on consumption behavior, mainly focusing on behavior that evokes positive emotions or behavior that evokes negative emotions, and less on the behavior that evokes both negative emotions and positive emotions (i.e., counterhedonic consumption) [16].

Based on Upward Comparison Theory [17] and Emotion Regulation Theory [18], this research showed that upward comparison leads to counterhedonic consumption. Previous research has shown that upward comparisons can awaken relative deprivation, a cognitive state in which individuals or groups perceive themselves to be at a disadvantage compared to a specific reference group, accompanied by negative emotions such as anger [19]. According to Emotion Regulation Theory, individuals seek methods to alleviate relative deprivation [18]. Counterhedonic consumption can help individuals alleviate relative deprivation [20,21,22]. Therefore, this research proposed that upward comparison can lead to counterhedonic consumption. In addition, previous research suggested that comparison targets (friend vs. stranger) can influence consumer behavior [23]. Thus, this research explored the moderating role of comparison targets in the effect of upward comparisons on counterhedonic consumption. Specifically, when the comparison target is a friend, an upward comparison (vs. non-upward comparison) leads to counterhedonic consumption. When the comparison target is a stranger, an upward comparison (vs. non-upward comparison) has no significant influence on counterhedonic consumption.

Five empirical studies were conducted in this research. Study 1 shows the effect of upward comparisons on backlash consumption. We then tested the mechanisms underlying relative deprivation through two different upward comparison scenarios (Studies 2 and 3). Study 4 tested the effect of upward comparisons on other consumption types to address whether upward comparisons lead to other consumption behaviors and not just counterhedonic consumption. Finally, Study 5 tests the moderating role of comparison targets between upward comparison and counterhedonic consumption.

This research makes three contributions. First, previous scholars examined how upward comparisons influence consumption behavior (e.g., impulse consumption, indulgent food consumption, compulsive consumption) that evokes negative emotions or consumption behavior (e.g., donation behavior, sustainable consumption) that evokes positive emotions [9,11,12,15]. To our knowledge, fewer scholars have focused on counterhedonic consumption that evokes both negative and positive emotions. This research adds to this area of the literature by examining the impact of upward comparisons in counterhedonic consumption. Second, this research extends the existing literature on the upward comparison in consumption by introducing a novel pathway (i.e., relative deprivation). To the best of our knowledge, our research is the first to examine the association between upward comparison and counterhedonic consumption and to demonstrate how upward comparisons affect consumers’ preferences for counterhedonic consumption. Third, this research complements the counterhedonic consumption literature by systematically investigating the distinction of comparison targets and showing that upward comparisons lead to counterhedonic consumption only when the comparison target is a friend.

In the next section, we review previous studies and formulate the hypotheses. Afterward, we report the results of five studies designed to test the hypotheses. Finally, we discuss the theoretical and practical contributions of this research, its limitations, and future research directions.

## 2. Literature Review and Research Hypothesis

### 2.1. Counterhedonic Consumption

Counterhedonic consumption refers to the consumption of products or services that not only elicit negative emotions such as fear and anxiety but also evoke positive emotions such as excitement, thrill, and pleasure in the individual [24]. For instance, sad experiences (e.g., sad music, sad novels, and sad movies) can make people enjoy them and feel pleasure that the “mind trumps the body” [25]. Horror-themed consumption (e.g., horror movies) not only awakens individuals’ negative emotions (e.g., fear) but also brings pleasure and enjoyment to individuals when it ends [26]. Extreme sports activate individuals’ fear and increase their perception of happiness in life [27].

Previous research has demonstrated that both the social environment and individual characteristics play important roles in influencing consumers’ preferences for counterhedonic consumption. For instance, in research on the social environment, Yang et al. (2022) found that resource scarcity influences consumers’ willingness to engage in counterhedonic consumption [24]. Consumers experiencing a diminished sense of control due to limited resources tend to exhibit a reduced preference for counterhedonic consumption. In the research on characteristics of individuals, Keinan and Kivetz (2011) discovered that consumers with achievement-orientated characteristics may purchase novel but unpleasant products or services (e.g., a stay in a freezer) to enrich their experiential experience [28]. Clasen et al. (2020) examined how intelligence and imagination positively predicted individuals’ preferences and frequency of horror consumption [29]. While scholars have investigated the impacts of individual characteristics and the social environment on counterhedonic consumption, to our knowledge, they were less focused on upward comparisons—subjective behavior that can influence consumers’ perceptions and thoughts [30,31].

### 2.2. Upward Comparison

Upward comparisons are prevalent in life and often harm an individual’s psychology [32]. Am upward comparison occurs when people compare themselves to someone that they perceive to be superior [5]. For instance, when people browse social media, they often see photos of joyful moments shared by friends or acquaintances [33]. If individuals perceive shortcomings in their own lives or appearance, they may experience dissatisfaction, thereby triggering feelings of inadequacy and insecurity [34]. In addition, within the workplace, individuals often engage in comparisons with colleagues regarding various aspects such as work achievements, the pace of promotion, or salary levels [35]. When individuals perceive themselves as falling behind others, they may experience feelings of inferiority and frustration, which can potentially impact their job performance and career advancement [36]. Likewise, comparisons with family members or friends during family gatherings or social events may elicit feelings of jealousy, dissatisfaction, and tension, disrupting family and social relationships and leading to conflict and estrangement [37].

Previous research has examined the effects of upward comparisons on consumption behavior, mainly focusing on behavior that evokes positive emotions or behavior that evokes negative emotions. In the research on consumption behavior that evokes negative emotions, Crusius and Mussweiler (2012) found that upward comparisons can lead to impulse consumption [6]. This consumption behavior can arouse negative emotions such as anxiety and regret [7]. In addition, Zheng et al. (2020) showed that upward comparisons can lead to compulsive consumption [9]. This consumption behavior also can arouse anxiety and regret [8]. Moreover, scholars have indicated that when individuals observe other people with thinner bodies than their own, they may feel inclined towards indulgent food consumption [11], which can awaken anxiety and guilt [10]. In research on consumption behavior that evokes positive emotions, Pak and Babiarz (2022) found that upward comparisons reduce individuals’ willingness to engage in charitable donations [12]. Aknin et al. (2017) showed that charitable donations can awaken pleasurable emotions [13]. In addition, Chen et al. (2024) found that participants in the upward comparison (vs. non-upward comparison) condition showed a lower willingness to engage in sustainable consumption [15]. Sustainable consumption can make consumers feel pleasure and improve their well-being [14]. Counterhedonic consumption is also prevalent in life, and it evokes negative emotions and activates positive emotions [24]. However, fewer scholars have directly examined the effect of upward comparison on counterhedonic consumption. Thus, this research addresses this gap.

### 2.3. Upward Comparison and Relative Deprivation

Upward comparisons typically can awaken individuals’ sense of unfairness [9]. People typically attribute the success of upward comparison targets to external factors rather than their own factors, which is accompanied by a strong sense of unfairness [38,39]. For instance, when employees discover that their colleagues with the same work experience as themselves are paid more than they are, they may perceive this as unfair treatment and consequently exhibit negative work attitudes [40]. In addition, in the ultimatum game, individuals tend to perceive a strong sense of unfairness when they realize they are being paid less than others, regardless of the actual amount they receive [41].

Relative deprivation refers to the cognitive state in which individuals or groups perceive themselves to be at a disadvantage compared to a specific reference group, which is often accompanied by negative emotions such as anger [11]. Previous research showed that relative deprivation stems from unfairness in upward comparison [42]. For instance, Cheung and Lucas (2016) found that individuals in lower social classes have a high relative deprivation and low well-being [43]. In addition, Osborne et al. (2013) showed that when a group suffers from unfair treatment, group members will have a high relative deprivation and show high protest intentions [44]. In conclusion, this research argues that upward comparisons awaken relative deprivation and influence consumer psychology and behavior.

### 2.4. Relative Deprivation and Counterhedonic Consumption

Relative deprivation is an emotional experience caused by upward comparisons [45]. In this experience, individuals experience negative emotions, such as anger and resentment, because they realize that others possess what they desire but they do not have [46,47]. According to Emotional Regulation Theory, individuals assess whether their mental state is good or bad, and subsequently take actions to minimize the difference between the two based on this assessment [18]. Relative deprivation undermines the individual’s mental state [48]. Thus, individuals will seek methods to alleviate relative deprivation.

Counterhedonic consumption is a vital consumption behavior that can alleviate relative deprivation. Counterhedonic consumption is the experience of both positive and negative emotions [24]. Engaging in these experiences of negative emotions (e.g., watching a horror movie and experiencing a haunted house) means the individual must face the challenge of overcoming the negative emotions. Previous research has shown that when individuals successfully overcome negative emotions, the individual’s confidence in their self-competence and value is enhanced, which raises self-esteem and arouses fulfillment [22]. This experience of fulfillment and self-esteem facilitates the individual’s ability to adjust their mental state and alleviate relative deprivation [20]. In addition, according to Consequence Modeling Theory, the relief that individuals feel at the end of a negative emotional experience helps to reduce the negative emotion (e.g., anxiety, frustration) and stimulate positive emotions (e.g., confidence, contentment, and optimism) [16]. The end of counterhedonic consumption means the end of negative emotions [24]. Therefore, counterhedonic consumption has an emotion-regulating function. The emotion-regulating function reduces the intensity of relative deprivation [21]. Based on this, the following hypothesis is proposed.

**Hypothesis 1.** 
*Compared to non-upward comparisons, upward comparisons lead to counterhedonic consumption.*


**Hypothesis 2.** 
*Relative deprivation mediates the effect of upward comparisons on counterhedonic consumption.*


### 2.5. Comparison Targets

Comparison targets refers to the target that the individual is referring to in the comparison process [23] and is an important factor that can influence individuals’ psychology in upward comparisons [49]. Friends and strangers are two target groups that individuals frequently compare. Previous research showed that individuals prefer to be friends with individuals with a high perceived similarity [50,51]. Thus, friends usually refer to others who share similar interests and social backgrounds with the individual [52]. In contrast, strangers in a social circle are usually others who have less interaction with the individual [53]. The reason for the low level of interaction is the differences between strangers and individuals [54]. Therefore, the difference in perceived similarity (the extent to which individuals perceive others to be similar to themselves) is one of the important distinctions between friends and strangers.

Upward comparisons awaken relative deprivation in an individual only when the comparison targets are similar to the individual. Yet, upward comparisons do not arouse relative deprivation when the comparison targets are not similar to the individual [55]. We expect comparison targets to moderate the effect of upward comparisons on relative deprivation and preferences for counterhedonic consumption. Specifically, upward comparisons arouse relative deprivation only when the comparison target is a friend, and thus lead to a preference for counterhedonic consumption. Therefore, we predict that:

**Hypothesis 3.** 
*Comparison targets moderate the effect of upward comparisons on counterhedonic consumption.*


**Hypothesis 3a.** 
*When the comparison target is a friend, upward comparison (vs. non-upward comparison) leads to more counterhedonic consumption.*


**Hypothesis 3b.** 
*When the comparison target is a stranger, the upward comparison (vs. non-upward comparison) has no significant influence on counterhedonic consumption.*


Our conceptual framework for the effect of upward comparison on counterhedonic consumption includes four constructs: upward comparison, relative deprivation, counterhedonic consumption, and comparison targets (see Figure 1).

## 3. Overview of Studies

We tested our hypotheses in five studies (see Table 1). Study 1 demonstrated that upward comparison (vs. non-upward comparison) leads to consumers’ preferences for counterhedonic consumption. The relative deprivation mediates our effect (Studies 2 and 3), and the effect disappears when the comparison target is a stranger, which reduces the individual’s perception of similarity to the comparison target, thus alleviating the individual’s relative deprivation (Study 5). In addition, Study 4 tested the effect of upward comparisons on other consumption types to address the question of whether upward comparisons lead to other consumption behaviors and not just counterhedonic consumption.

## 4. Study 1

Study 1 tested whether upward comparison (vs. non-upward comparison) leads to counterhedonic consumption (H1). In addition, since employment is also one of the students’ concerns [56], Study 1 chose the employment situation as a comparison scenario. Moreover, Study 1 also aimed to rule out the sense of security as an alternative explanation.

### 4.1. Design and Participants

Study 1 was conducted from 17–19 February 2023. Since MBA student groups are concerned with employment situations [56], the selection of MBA students as subjects is beneficial for studying upward comparisons. Thus, this study recruited 390 MBA students through offline recruitment from a university in southern China. Before the study began, participants were told that they would participate in a survey on horror houses. Each participant in the study will receive CNY 5 as a reward. We excluded 6 participants who failed the attention check questions, leaving 384 valid participants (M_age_ = 29.64 years, 65.90% females). Participants were randomly assigned to a (social comparison: upward comparison vs. downward comparison vs. control) between-subjects design.

### 4.2. Procedure

Social comparison. Participants were asked to imagine that their internship experience was the situation we described (“I had one–two internships in small companies”, “I got an offer from a small company”). Then, participants were asked to read different character materials to activate the comparison process. Specifically, in the upward comparison condition, the comparison target was described as an excellent classmate (“He/She was employed in a position with a monthly salary exceeding CNY 20,000”, “He/She received multiple offers from prestigious companies”). In the non-upward comparison condition, the comparison target was described as a classmate who has a difficult employment situation (“He/She had no internship experience”, “He/She did not receive any offers from companies”). In the control condition, the comparison target was described as an average classmate (“He/She had one-two internship experiences in small companies”, “He/She got an offer from a small company”). Next, according to the research of Zheng et al. (2018), participants responded to a manipulation check question, which asked how they felt about their employment situation as compared to the comparison target (1 = very much worse off, 7 = very much better off) [57].

Sense of security. According to the research of Yang et al. (2022), for the sense of security, participants indicated the extent to which they felt they had a sense of security at the moment (1 = very much lack a sense of security, 7 = very much have a sense of security) [24].

Counterhedonic consumption. Participants were shown a haunted house poster along with the description: “This decaying century-old mansion holds many unknown secrets. It’s a murderer’s party, a burglar’s lair, and there are even rumors of other mysteries hidden inside this dark, decaying, old mansion”. According to the research of Yang et al. (2022), participants indicated the extent to which they would enjoy this haunted house experience on a seven-point scale (1 = would not enjoy at all, 7 = would very much enjoy) [24]. Participants completed demographic measures.

### 4.3. Results

Manipulation checks. An ANOVA on the manipulation check question revealed a significant influence of social comparison on the evaluation of the comparison target (F(2, 381) = 100.23, *p* < 0.001, η^2^ = 0.34). Participants in the upward comparison condition (M_upward comparison_ = 4.89) exhibited higher evaluations of the comparison target than participants in the downward comparison group (M_downward comparison_ = 2.25; F(1, 256) = 191.47, *p* < 0.001, η^2^ = 0.43) and participants in the control group (M_control_ = 3.14; F(1, 251) = 80.81, *p* < 0.001, η^2^ = 0.24). In addition, participants in the downward comparison condition had lower evaluations of the comparison target than participants in the control group (F(1, 256) = 22.51, *p* < 0.001, η^2^ = 0.081). The manipulation of the upward comparison is successful.

Anticipated enjoyment. A one-way analysis of variance (ANOVA) revealed a significant effect of social comparison on the anticipated enjoyment of the haunted house (F(2, 381) = 6.507, *p* = 0.002, η^2^ = 0.03). Compared to participants in the control group (M_control_ = 3.94; F(1, 251) = 10.05, *p* = 0.002, η^2^ = 0.03) and participants in the downward comparison group (M_downward comparison_ = 3.91; F(1, 255) = 10.27, *p* = 0.002, η^2^ = 0.039), participants in the upward comparison condition (M_upward comparison_ = 4.61) exhibited a higher anticipated enjoyment for the haunted house. There was no significant difference in the enjoyment of haunted houses between participants in the control group and the downward comparison group (F(1, 256) = 0.02, *p* = 0.89, η^2^ < 0.001). This result supported H1.

Sense of security. A one-way analysis of variance (ANOVA) revealed no significant effect of social comparison on the sense of security (F(2, 381) = 0.32, *p* = 0.72, η^2^ = 0.02). The sense of security with participants in the upward comparison condition (M_upward comparison_ = 5.34) was not significantly different from participants in the control condition (M_control_ = 5.19; F(1, 251) = 0.72, *p* = 0.39, η^2^ = 0.001) and participants in the downward comparison condition (M_downward comparison_ = 5.27; F(1, 255) = 0.14, *p* = 0.70, η^2^ = 0.003). There was also no significant difference in the sense of security between participants in the control condition and participants in the downward comparison condition (F(2, 258) = 0.17, *p* = 0.67, η^2^ = 0.001).

Mediated analysis. To test whether upward comparisons affected participants’ sense of security, we performed a mediation analysis (5000 bootstraps; PROCESS Model 4; Hayes 2017) with sense of security as a mediator. The analysis revealed that the effect of the upward comparison on the sense of security was not significant (indirect effect = −0.016, SE = 0.020, 95% CI = [−0.061, 0.020]). Thus, we ruled out alternative explanations for the sense of security.

### 4.4. Discussion

Study 1 demonstrated that participants in the upward comparison (vs. a non-upward comparison) condition showed higher preferences for counterhedonic consumption, supporting H1. In addition, Study 1 rules out the competing explanation for the sense of security. However, the haunted house is a service experience and it is more realistic than other counterhedonic consumption (e.g., horror games and horror movies) [58]. The high authenticity may influence our result. To exclude the inference of authenticity and improve the robustness of our study, Study 2 chose a comparative scenario of the socioeconomic status of employees and used a horror game as a stimulus.

## 5. Study 2

Study 2 tested whether relative deprivation mediates the effect of upward comparisons on counterhedonic consumption (H2). To exclude authenticity and improve the robustness of our study, we chose a comparative scenario of the socioeconomic status of employees and used a horror game as a stimulus. In addition, Study 2 investigated another closely related psychological construct—self-efficacy. Specifically, prior research suggests that upward comparisons might influence self-efficacy [59]. It might be argued that the effect of upward comparisons on counterhedonic consumption is partly driven by self-efficacy. Thus, we also measured this construct and examined its potential underlying role.

### 5.1. Design and Participants

Study 2 was conducted from 1–4 March 2023. The comparison scenario chosen for Study 2 is the comparison of socioeconomic status. Employees are typically more concerned about socioeconomic status compared to other groups. The selection of employees as subjects is beneficial for activating upward comparisons in Study 2. A total of 210 employees from a medium-sized company in southern China were recruited offline to participate in this study in exchange for CNY 5 as a reward. We excluded 2 participants who failed the attention check questions, leaving 208 valid participants (Mage = 30 years, 56.73% females). Before Study 2 began, participants were told that they would participate in a survey on horror games. Participants were randomly assigned to a (upward comparison: yes vs. no) between-subjects design.

### 5.2. Procedure

Upward comparison. Participants were asked to imagine that their socioeconomic status was the situation we described (“My job performance is not excellent”, “Like most people, I had missed many opportunities for advancement”). Then, participants were asked to read different character materials to activate the comparison process. Specifically, in the upward comparison condition, the comparison target was described as an excellent colleague (“He/She had received an opportunity to pursue further education abroad”, “He/She had been promoted to a senior management position”). In the non-upward comparison condition, the comparison target was described as an average colleague (“His/Her job performance is not excellent”, “Like most people, he/she had missed many opportunities for advancement”). Next, according to the research of Zheng et al. (2018) [57], participants responded to a manipulation check question, which asked how they felt about their socioeconomic status as compared to the comparison target (1 = very much worse off, 7 = very much better off).

Relative deprivation. According to the research of Callan et al. (2011), participants were asked to complete the relative deprivation scale (see Appendix A) [60].

Self-efficacy. According to the research of Chen et al. (2001), participants were then asked to complete the self-efficacy scale (see Appendix A) [61].

Counterhedonic consumption. Participants were shown a poster of a horror game, *Trials*, along with a short description: “*Trials* is a horror game for Steam that immerses players in a world filled with terror and ominousness, with the sole objective being to survive”. Referring to the research of Yang et al. (2022), participants were asked to indicate the extent to which they preferred playing this horror game on a Likert seven-point scale (1 = dislike very much, 7 = like very much) [24]. Participants were asked to provide demographic information.

### 5.3. Results

Manipulation check. A one-way analysis of variance (ANOVA) revealed a significant effect of upward comparisons on the evaluation of the comparison target (M_upward comparison_ = 5.28 vs. M_non-upward comparison_ = 2.55; F(1, 206) = 210.88, *p* < 0.001, η^2^ = 0.51). Our manipulation of the upward comparison was successful.

Preference. A one-way analysis of variance (ANOVA) revealed a significant effect of upward comparisons on the preference for the horror game (F(1, 206) = 4.84, *p* = 0.03, η^2^ = 0.02). Compared to participants in the non-upward comparison condition (M_non-upward comparison_ = 4.46), participants in the upward comparison condition (M_upward comparison_ = 4.96) showed a high preference for the horror game. This result proved H1.

Relative deprivation. A one-way analysis of variance (ANOVA) revealed a significant effect of upward comparisons on relative deprivation (M_upward comparison_ = 4.07 vs. M_non-upward comparison_ = 3.37; F(1, 206) = 21.40, *p* < 0.001. η^2^ = 0.09).

Self-efficacy. A one-way analysis of variance (ANOVA) did not show a significant influence of upward comparisons on self-efficacy (M_upward comparison_ = 5.83 vs. M_non-upward comparison_ = 5.69; F(1, 206) = 1.36, *p* = 0.24. η^2^ = 0.07).

Mediated analysis. To test whether upward comparisons affected participants’ preferences for the horror game through relative deprivation, we performed a mediation analysis (5000 bootstraps; PROCESS Model 4; Hayes 2017) with relative deprivation as a mediator. The analysis revealed that the relative deprivation significantly mediated the effect of the upward comparison on the preference for the horror game (indirect effect = −0.185, SE = 0.084, 95% CI = [−0.337, −0.040]), but the direct effect of the upward comparison on the preference for the horror game was not significant (indirect effect = −0.304, SE = 0.230, 95% CI = [−0.759, 0.150]). These results proved H2. In addition, we also measured the mediating role of self-efficacy through the model described above. The results showed that the mediating role of self-efficacy was not significant (indirect effect = 0.075, SE = 0.070, 95% CI = [−0.041, 0.238]).

### 5.4. Discussion

Study 2 demonstrated that upward comparisons can increase individuals’ preferences for horror games, supporting H1. In addition, Study 2 demonstrated the mediating role of relative deprivation in the effect of upward comparisons on counterhedonic consumption, supporting H2. Moreover, Study 2 ruled out competing explanations for self-efficacy. However, there are still some shortcomings in the above studies. Studies 1 and 2 used haunted houses and horror-themed games as stimuli to measure counterhedonic consumption. However, both the haunted house and the game were interactive, which may have influenced the results [62]. Study 3 excluded the effect of interactivity through the use of horror movies. In addition, prior research suggested that familiarity may influence evaluations [63]. To eliminate the influence of familiarity, Study 3 chose a virtual movie as the stimulus.

## 6. Study 3

Previous studies have demonstrated H1 and H2 in different populations. To improve the robustness of the research, Study 3 chose another direction of comparison for the student population: academic achievement [64]. In addition, Study 3 chose a virtual movie as the stimulus to exclude the interference of familiarity and intimacy. Moreover, previous research has shown that evaluation can reflect preferences [65]. Thus, to improve the validity of the studies, Study 3 added a measure of evaluation of counterhedonic consumption.

### 6.1. Design and Participants

Study 3 was conducted from 7–11 March 2023. The comparison scenario chosen for Study 3 is the comparison of academic achievement. Since MBA student groups are concerned with academic achievement [64], the selection of MBA students as subjects is beneficial for activating upward comparisons. A total of 145 MBA students from a university in southern China participated in Study 3 in exchange for course credit through offline recruitment. We excluded 7 participants who failed the attention check questions, leaving 138 valid participants (M_age_ = 30.58 years, 57.80% females). Participants were randomly assigned to a (upward comparison: yes vs. no) between-subjects design.

### 6.2. Procedure

Upward comparison. Participants were asked to imagine that their academic experience was the situation we described (“I have not received academic honors”, “Like most students, my grades were in the lower middle of his/her major”). Then, participants were asked to read different character materials to activate the comparison process. Specifically, in the upward comparison condition, the comparison target was described as an excellent classmate (“He/She got multiple first-class scholarships”, “His/Her grades located at the Top 5% overall in your major”). In the non-upward comparison condition, the comparison target was described as an average classmate (“He/She has not received academic honors”, “Like most students, his/her grades were in the lower middle of his/her major”). Referring to the research of Zheng et al. (2018), participants were asked a manipulation check question, which asked how they felt about their academic achievement as compared to the comparison target (1 = very much worse off and 7 = very much better off) [57].

Relative deprivation. Participants were asked to complete a relative deprivation scale, the same as Study 2 (see Appendix A).

Counterhedonic consumption. Participants were shown a poster of a horror movie, *Last Supper*, along with a short description: “In a century-old mansion located in the wilderness, individuals indulge in a lavish meal. However, something peculiar is lurking outside the window, and white bones scuttle across the ground”. According to the research of Maheswaran et al. (1992), participants were asked to indicate their evaluation scale on the horror movie (see Appendix A). Participants were asked to provide demographic information [66].

### 6.3. Results

Manipulation check. A one-way analysis of variance (ANOVA) revealed a significant effect of upward comparison on the evaluation of the comparison targets (M_upward comparison_ = 5.37 vs. M_non-upward comparison_ = 2.15; F(1, 136) = 83.82, *p* < 0.001, η^2^ = 0.38). Our manipulation of the upward comparison was successful.

Evaluation. Analysis of variance (ANOVA) revealed the significant effects of upward comparison on the evaluation of horror films (F(1, 136) = 6.61, *p* = 0.011, η^2^ = 0.046). Participants in the upward comparison condition (M_upward comparison_ = 4.84) had higher evaluations than those in the non-upward comparison condition (M_non-upward comparison_ = 4.19), supporting H1.

Relative deprivation. Analysis of variance (ANOVA) revealed the significant effects of upward comparison on the relative deprivation (F(1, 136) = 8.15, *p* = 0.005, η^2^ = 0.05). Participants in the upward comparison condition (M_upward comparison_ = 4.14) had higher relative deprivation than those in the non-upward comparison condition (M_non-upward comparison_ = 3.55).

Mediation analysis. To test whether upward comparisons affected participants’ evaluation of horror movies through relative deprivation, we performed a mediation analysis (5,000 bootstraps; PROCESS Model 4; Hayes 2017) with relative deprivation as a simultaneous mediator. The analysis revealed that the relative deprivation significantly mediated the effect of the upward comparison on the evaluation of the horror movie (indirect effect = 0.188, SE = 0.086, 95% CI = [0.041, 0.381]), supporting H2.

### 6.4. Discussion

Study 3 supported the mediating effect of the relative deprivation underlying the influence of the upward comparison on counterhedonic consumption again, supporting H2. In addition, Study 3 showed that upward comparisons increase evaluations of the horror movie, supporting H1. Moreover, Study 3 excluded the interference of familiarity and intimacy. However, we previously activated upward comparisons by reading material about comparison targets in Studies 1-3. Upward comparisons can be activated by viewing the content of others’ social media profiles. In Study 4, we activated the upward comparison by viewing screenshots of others’ social media profiles. In addition, Studies 1-3 demonstrated that upward comparison increases counterhedonic consumption. However, does upward comparison lead to any other types of consumption? To address this question, Study 4 tested the effect of upward comparisons on other consumption types.

## 7. Study 4

Study 4 tested the effect of upward comparisons on other consumption types to address whether upward comparisons lead to other consumption behaviors and not just counterhedonic consumption. In Study 4, we also activate the upward comparison of participants by asking them to view screenshots of other people’s social media profiles.

### 7.1. Design and Participants

Study 4 was conducted from 27 March to 29 March 2023. To improve the external validity of the studies, 210 participants were recruited online from the Credamo platform (a data collection platform similar to Mturk) to participate in this study. Participants would receive CNY 5 as a reward. We excluded 2 participants who failed the attention check questions, leaving 208 participants (M_age_ = 30.57 years, 66.34% females) for the analyses. Participants were randomly assigned to a 2 (upward comparison: yes vs. no) × 2 (consumption type: counterhedonic consumption vs. control) between-subjects design.

### 7.2. Procedure

Upward comparison. Participants were asked to imagine that their academic experience was the situation we described (“Meet the school’s graduation requirements” and “Never won a scholarship, but lived a happy life”). Then, to activate the comparison process, participants were asked to browse different screenshots of other people’s social media profiles. Specifically, in the upward comparison condition, participants lookes at a screenshot to learn the account owner’s outstanding academic achievements, such as “multiple first-class scholarships” and “multiple honors for outstanding students”. In the non-upward comparison condition, participants viewed the screenshot to learn the account owner’s general academic achievements, such as “Meet the school’s graduation requirements” and “Never won a scholarship, but lived a happy life”. Referring to the research of Zheng et al. (2018), participants were asked a manipulation check question, which asked how they felt about their academic achievement as compared to the comparison target (1 = very much worse off and 7 = very much better off) [57].

Choices. Referring to the research of Yang et al. (2022), in the counterhedonic consumption condition, participants were shown the poster of *Final Destination* (a horror movie) [24]. In the control condition, participants were shown the poster of *Free Solo* (a documentary film). The two movies had similar release dates, and they have similar IMDB profiles (IMDB is an online database with information on movie actors, movies, TV shows, TV stars, and movie productions). Afterward, participants were asked to tell us whether they would like to watch the movie after the experiment was over. Finally, participants were required to provide demographic information.

### 7.3. Results

Manipulation tests. A one-way analysis of variance (ANOVA) revealed a significant effect of upward comparison on the evaluation of comparison targets. We found that participants in the upward comparison had a significantly higher evaluation of the comparison target compared to participants in the non-upward comparison (M_upward comparison_ = 5.31 vs. M_non-upward comparison_ = 2.63; F (1, 206) = 222.20, *p* < 0.001, η^2^ = 0.51).

Choice. A logistic regression analysis of choice (Final Destination vs. Free Solo) yielded the upward comparison × consumption-type interaction (β = −0.60, Wald χ^2^(1, N = 208) = 5.32, *p* = 0.021). Specifically, in the counterhedonic consumption condition, participants in the upward comparison condition (71.11%) chose horror movies at a higher rate than participants in the non-upward comparison condition (48.93%; χ^2^(1, N = 92) = 7.50, *p* = 0.006). However, in the control condition, there was no significant difference in the proportion of participants in the non-upward comparison condition (51.92%) who chose a documentary movie versus those in the upward comparison condition (51.56%; χ^2^(1, N = 116) = 0.062, *p* = 0.803). These results support H1 and rule out a competitive explanation that upward comparisons lead to all consumption behaviors (see Figure 2).

### 7.4. Discussion

Study 4 showed that the interaction of upward comparison significantly affected movie preferences (H1). In addition, Study 4 also addressed the question of whether upward comparisons lead to other consumption behaviors and not just counterhedonic consumption. In Study 5, we tested the interactive effect of upward comparison and comparison targets on counterhedonic consumption, and relative deprivation mediates the effect of upward comparison and comparison targets on counterhedonic consumption.

## 8. Study 5

Study 5 examined whether the comparison targets moderate the effect of upward comparisons on counterhedonic consumption.

### 8.1. Design and Participants

Study 5 lasted for 3 days, from 27–29 March 2023. A total of 180 participants were recruited online from the *Credamo* platform to participate in this study. Participants would receive CNY 5 as a reward. We excluded 7 participants who failed the attention check questions, leaving 173 participants for the analyses (M_age_ = 30.07 years, 65.90% females). Participants were randomly assigned to a 2 (upward comparison: yes vs. no) × 2 (comparison target: stranger vs. friend) between-subjects design.

### 8.2. Procedure

Comparison targets. According to the research of Zhan et al. (2018), when the comparison target was a friend, participants were asked to recall a friend and to write down their name [67]. When the comparison target was a stranger, participants were asked to recall a stranger around them and to write down their name. Afterward, participants were asked to imagine that they and their friend or the stranger had achieved the following academic achievements.

Upward comparison. Participants were asked to imagine that their academic experience was the situation (“I have not received academic honors”, “Like most students, my grades were in the lower middle of my major”). Then, participants were asked to read different character materials to activate the comparison process. Specifically, in the upward comparison condition, the comparison target was described as an excellent student (“He/She got multiple first-class scholarships”, “His/Her grades are located in the top 5% overall in your major”). In the non-upward comparison condition, the comparison target was described as an average student (“He/Her has not received academic honors”, “Like most students, his/her grades were in the lower middle range of his/her major”). Referring to the research of Zheng et al. (2018) [57], participants were asked a manipulation check question, which asked how they felt about their academic achievement as compared to the comparison target (1 = I am much worse off and 7 = I am much better off).

Relative deprivation. According to the research of Callan et al. (2011), participants were asked to complete the relative deprivation scale, the same as in Study 2 (see Appendix A) [60].

Evaluation. Participants were shown a poster of a horror movie, *The Haunting of the Morgue 2: The Georgia Haunted House*, along with a short description: “A family moves into a centuries-old mansion in the wilderness. There, an unknown evil force stirs. Headless corpses, midnight apparitions, and friends who exist only in the imagination make the family experience a terrible ordeal”. Afterward, according to the research of Maheswaran et al. (1992), participants were asked to complete the evaluation scale on the horror movie (see Appendix A) [66]. Finally, participants were asked to provide demographic information.

### 8.3. Results

Manipulation tests. An ANOVA on the manipulation check question confirmed that compared to those in the non-upward comparison condition, participants in the upward comparison condition indeed had a higher evaluation of the comparison targets (M_upward comparison_ = 5.35 vs. M_non-upward comparison_ = 2.58; F(1, 172) = 243.86, *p* < 0.001, η^2^ = 0.58). The manipulations of upward comparison were successful.

Evaluation. A two-way ANOVA on the evaluation of horror movies revealed a significant interaction effect of comparison targets and upward comparison (F(1, 169) = 4.68, *p* = 0.03, η^2^ = 0.02). Specifically, when the comparison target was a friend, participants in the upward comparison (vs. non-upward comparison) condition had higher evaluations of horror movies (M_upward comparison_ = 5.58 vs. M_non-upward comparison_ = 4.61; F(1, 74) = 14.06, *p* < 0.001, η^2^ = 0.16). In contrast, when the comparison target was a stranger, the effect of upward comparison (vs. non-upward comparison) on preferences for a horror movie was not significant (M_upward comparison_ = 4.69 vs. M_non-upward comparison_ = 4.66; F(1, 96) = 0.006, *p* = 0.938, η^2^ < 0.001). These results proved H3 (see Figure 3).

Relative deprivation. A two-way ANOVA revealed a significant interaction effect of comparison targets and upward comparisons on relative deprivation (F(1, 169) = 9.89, *p* = 0.002, η^2^ = 0.05). Specifically, when the individual comparison target was a friend, participants in the upward comparison (vs. non-upward comparison) had a higher relative deprivation (M_upward comparison_ = 4.69 vs. M_non-upward comparison_ = 3.95; F(1, 74) = 8.75, *p* = 0.004, η^2^ = 0.11). In contrast, when the individual comparison target was a stranger, the effect of upward comparison (vs. non-upward comparison) on relative deprivation was not significant (M_upward comparison_ = 3.61 vs. M_non-upward comparison_ = 3.85; F(1, 95) = 1.52, *p* = 0.22, η^2^ = 0.018).

Moderated mediation. A moderated mediation test (Hayes and Preacher 2014, model 7) with the upward comparison as the independent variable, evaluations of horror movies as the dependent variable, relative deprivation as the mediating variable, and the comparison targets as the moderating variable was significant (Index = −0.326, SE = 0.164, 95%CI = [−0.708, −0.074]). In further analysis, when the individual comparison target was a friend, relative deprivation mediated the effect of upward comparison on the evaluation of the horror movie (indirect effect = 0.246, SE = 0.124, 95%CI = [0.053, 0.525]). However, when the individual comparison target was a stranger, the mediating effect was not significant (indirect effect = −0.079, SE = 0.073, 95%CI = [−0.246, 0.042]). These findings proved H3.

### 8.4. Discussion

Study 5 showed that comparison targets moderate the effect of upward comparison on counterhedonic consumption by relative deprivation, which supports H3. Especially when the comparison target is a friend, upward comparisons can arouse relative deprivation and thus lead to counterhedonic consumption. However, when the comparison target is a stranger, upward comparisons (vs. non-upward comparison) have no significant influence on relative deprivation and counterhedonic consumption.

## 9. General Discussion

Previous research mainly focused on the effects of upward comparisons on consumer behavior that can arouse negative emotions (e.g., impulse consumption, indulgent food consumption, compulsive consumption) or consumer behavior that can arouse positive emotions (e.g., donation behavior, sustainable consumption); to our knowledge, consumer behavior that can arouse both negative and positive emotions has been studied less. Therefore, this paper explores the effect of upward comparisons on counterhedonic consumption. Across five studies, we found that upward comparisons lead to counterhedonic consumption. Moreover, relative deprivation mediates the effect of the upward comparison on counterhedonic consumption. We also revealed that the effect disappears when the comparison target is a stranger, as upward comparisons with strangers fail to arouse relative deprivation in individuals. Study 1 used a haunted house as an experiential stimulus to evaluate participants’ attitudes toward counterhedonic consumption. Compared to other counterhedonic consumptions (e.g., horror games, horror movies), this immersive experience of a haunted house is perceived to have a higher degree of authenticity. In Study 2, we used a horror game as a stimulus, excluding the interference of authenticity. Haunted houses and horror games are usually counterhedonic consumptions in which multiple people participate together and interact. To mitigate the potential interference of interactivity, Study 3 used horror movies as stimuli. The findings from Studies 1–3 highlight the effect of upward comparisons on counterhedonic consumption. Upward comparisons may influence all consumption behaviors, not only counterhedonic consumption. Study 4 endeavored to address this question by examining the effect of upward comparisons on other types of consumption behaviors. Study 5 demonstrated the moderating effect of comparison targets. Specifically, when the comparison target was a friend, upward comparisons led to counterhedonic consumption. When the comparison target was a stranger, upward comparisons (vs. non-upward comparisons) did not significantly influence counterhedonic consumption.

### 9.1. Theoretical Implication

First, this research fills a gap in the upward comparison literature by identifying how upward comparisons can influence the preference for counterhedonic consumption. Upward comparisons are a critical factor in marketing and sales success [68], although previous research has examined the effect of upward comparisons on consumption behavior that can evoke either negative or positive emotions, such as impulse consumption [6], indulgent food consumption [11], compulsive consumption [9], donation behavior [12], and sustainable consumption [15]. However, few scholars have focused on counterhedonic consumption that can evoke positive and negative emotions. To the best of our knowledge, this research is the first to directly examine the association between upward comparisons and counterhedonic consumption and to demonstrate how upward comparison influences counterhedonic consumption.

Second, this research also extends the existing literature on relative deprivation in individual behavior by identifying relative deprivation as a novel pathway that underlies the impact of upward comparisons on counterhedonic consumption. Previous research has explored the effects of relative deprivation on dangerous behaviors that are harmful to individuals and found that unfairness can lead to relative deprivation and increase the likelihood that consumers engage in dangerous behaviors, such as smoking, alcohol abuse, and gambling [69,70,71]. Counterhedonic consumption is conceptually distinct from dangerous behaviors. Dangerous behaviors are categorically harmful experiences, and counterhedonic consumption is a genre of entertainment commonly sought for enjoyment. This research focuses on people’s preferences for entertainment options in a counterhedonic consumption domain.

Finally, this research enriches the literature on comparison targets (friends vs. strangers) in upward comparisons. Previous research has explored the effects of upward comparison targets (friends vs. strangers) on individual psychology in competitive relationships. In simple gambling tasks, a stranger’s success leads individuals to show higher negative emotions and lower satisfaction compared to a friend’s success [72]. However, this research explored the effects of upward comparison targets (friends vs. strangers) on individual psychology in noncompetitive relationships and observed different results. In noncompetitive relationships, upward comparisons with friends (vs. strangers) led individuals to develop higher relative deprivation and show more negative emotions.

### 9.2. Practical Implications

First, in recent years, counterhedonic consumption has been popular among consumers, yet we know very little about the factors that influence individual preferences for counterhedonic consumption. Upward comparison is a common phenomenon that can influence the psychological state of consumers, thereby affecting their behavior. This research indicated that upward comparisons increase consumers’ preference for counterhedonic consumption. Our research findings suggest that entrepreneurs in the start-up preparation stage can decide whether to engage in business activities related to counterhedonic consumption by considering the upward comparison of consumers in the market. In addition, our research also reminds companies related to counterhedonic consumption to market their products to consumer groups that are prone to upward comparisons, such as students, etc.

Second, this research found that alleviating relative deprivation is an important motivation for consumers to choose counterhedonic consumption. Hence, providers of counterhedonic products and services should recognize the importance of designing products or services that are beneficial to alleviate relative deprivation and maintain mental health. For instance, while enhancing the entertainment and horror of games, horror-themed parks should also focus on the healing nature of the game, setting some healing solutions for certain plot games, or examine how to help individuals alleviate relative deprivation through horror games. This not only attracts consumers and improves the profitability of the firm, but also benefits the mental health of consumers.

Finally, understanding the relationship between upward comparisons and counterhedonic consumption has important implications for individuals’ psychological and mental health. Scholars have found that upward comparison can lead to preferences for consumption behaviors, which is harmful to individuals. For instance, upward comparison can lead to impulse consumption [6], which reduces emotional health and self-esteem [73]. In addition, upward comparisons are positively associated with compulsive consumption, which may arouse negative emotions (e.g., anxiety and regret) and have a detrimental effect on mental health [8,74]. Moreover, upward comparison can lead to indulgent food consumption, which can awaken negative emotions (e.g., anxiety and guilt) and damage consumers’ physiological health and well-being [10,75,76]. Our research reminds consumers that they can choose counterhedonic consumption to mitigate the relative deprivation of upward comparisons and its adverse effects [77,78,79,80,81].

### 9.3. Limitations and Future Research

Our findings suggest several directions for future research. First, this paper mainly focused on one of the most important types of counterhedonic consumption—horror consumption. Scholars can examine whether upward comparisons influence other types of counterhedonic consumption [82,83]. For example, upward comparisons might also influence the consumers’ preference for sadness consumption (e.g., sad music, tragedy novels), as consumers’ relative deprivation might help them cope with sad emotions.

Second, this research explored the mediating mechanism of relative deprivation and excluded some alternative mediators, such as a sense of security and self-efficacy. Future research could explore other possible mediating mechanisms, such as a sense of control, self-esteem, and stress [84,85]. Thus, upward comparisons (vs. non-upward comparisons) may increase an individual’s willingness to purchase various products.

Finally, we also encourage future research to explore other boundary conditions of the positive effect of upward comparisons on counterhedonic consumption. Our research shows that the effect occurs because upward comparisons arouse relative deprivation, and the effect disappears when the comparison target is a stranger. However, in competitive relationships, the success of strangers may also awaken an individual’s relative deprivation. For example, when competing for the same position, the victory of unfamiliar candidates may lead to strong relative deprivation among individuals [86]. Future research could examine how different comparison targets in competitive relationships moderate our effect. In addition, future research could investigate whether the positive effect of upward comparison on counterhedonic consumption is attenuated when the comparison targets are friends with different relationship strengths.

## Figures and Tables

**Figure 1 behavsci-14-00374-f001:**
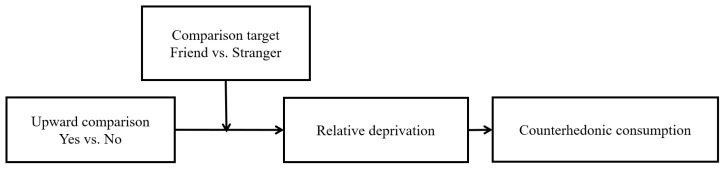
Theoretical model.

**Figure 2 behavsci-14-00374-f002:**
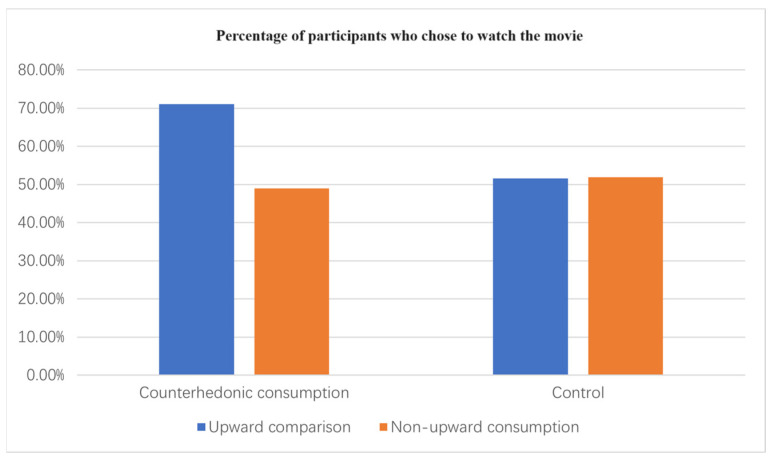
The interactive effect of upward comparisons and consumption type.

**Figure 3 behavsci-14-00374-f003:**
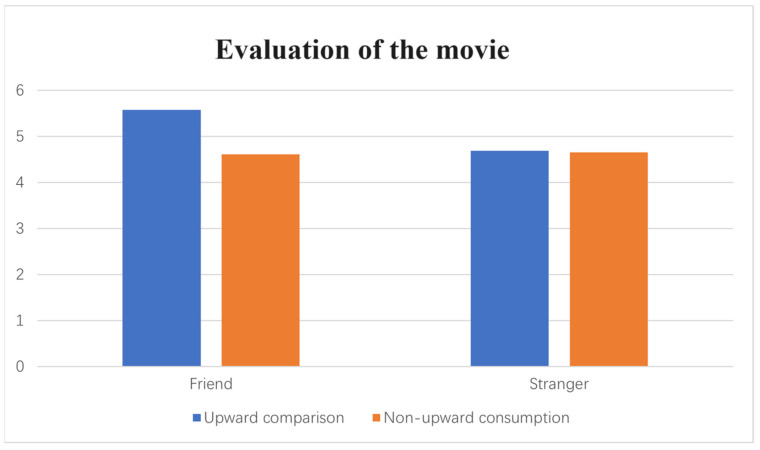
The interactive effect of upward comparisons and comparison targets.

**Table 1 behavsci-14-00374-t001:** Overview of studies.

Overview of Studies					
	Participants	Design	Independent Variable	Dependent Variable	Main Findings
Study 1	384 participants (M_age_ = 29.64 years, 257 females)	One-factor three-level (social comparison: upward comparison vs. downward comparison vs. control) between-subjects design	Upward comparison vs. downward comparison vs. control comparison scenario: employment situation of students	Anticipated enjoyment consumption type: haunted house	(1) Verifying H1 (2) Ruling out the sense of security
Study 2	208 participants (M_age_ = 30 years, 118 females)	One-factor two-level (upward comparison: yes vs. no) between-subjects design	Upward comparison vs. non-upward comparison comparison scenario: Socioeconomic status of employees	Preference consumption type: horror game	(1) Verifying H1 and H2 (2) Ruling out self-efficacy and authenticity of the haunted house
Study 3	138 participants (M_age_ = 30.58 years, 80 females)	One-factor two-level (upward comparison: yes vs. no) between-subjects design	Upward comparison vs. non-upward comparison comparison scenario: academic achievement of students	Evaluation consumption type: horror movie	(1) Verifying H1 and H2 (2) Excluding alternative explanations for familiarity and intimacy
Study 4	208 participants (M_age_ = 30.57 years, 138 females)	2 (upward comparison: yes vs. no) × 2 (consumption type: counterhedonic consumption vs. control) between-subjects design	Upward comparison vs. non-upward comparison comparison scenario: academic experience of students	Choice consumption type: horror movie or documentary movie	(1) Verifying H1 and H2 (2) Addressing the question of whether upward comparisons lead to other consumption behaviors and not just counterhedonic consumption.
Study 5	173 participants (M_age_ = 30.07 years, 114 females)	2 (upward comparison: yes vs. no) × 2 (comparison targets: stranger vs. friend) between-subjects design	Upward comparison vs. non-upward comparison comparison scenario: academic experience of students	Evaluation consumption type: horror movie	(1) Verifying H3 (2) Demonstrating the moderating, mediating role of comparison targets

## Data Availability

The data presented in this study are available on request from the corresponding author. The data are not publicly available due to the need to maintain the confidentiality of study participants.

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
