# Peer review of "Negative Emotions Will Be Welcomed: The Effect of Upward Comparison on Counterhedonic Consumption"

_behavsci, 2024, doi:10.3390/bs14050374_

Round 1

Reviewer 1 Report

Comments and Suggestions for Authors

Paper is written as report on done activities and findings and not as scientific paper with respect to academic style.

More information of novelty has to be described and clearly stated as for every step done by researchers there are references to other authors and previous publications.

Are research data representative if “….students from a university in southern China participated in Study 3 in exchange for credit” (lines 403 and 404); other selection of participants also have to be more precise written. More explanations of representative sample have to be added.

Not clear, what authors thought by “Participants were randomly assigned to one of two conditions” (line 329; line 406 and 407)Participants were randomly assigned to one of four conditions” (line 474; 531 and 532) and other places as there is no additional explanation.

Comments on the Quality of English Language

Some explanations have to be added. More academic style has to be applied instead of paper as report on done research.

Reviewer 2 Report

Comments and Suggestions for Authors

Thank you for the opportunity to review your manuscript. Please my comments below.

I think the concepts of counter hedonic consumption and upward comparison are very interesting. In the introduction, I would suggest linking them together. It is like you have two different topics based on the first two paragraphs. How do these concepts intercept? Horror and social media.. wanting to engage because of social media?

Great introduction of theories, but I would have liked a brief introduction of the studies conducted just to provide extra details and justification.

The visual framework is tough to see. I would recommend updating it to better viewing.

An introduction paragraph is needed in the methods section to introduce the overall methodology. Again, there are lots of studies conducted and justification is needed.

How were the participants recruited in study 1? Was a survey used? Paper or online? What software was used?

How were the participants recruited in study 2? Was a survey used? Paper or online? What software was used?

How were the participants recruited in study 3? Was a survey used? Paper or online? What software was used?

What is Credamo?

Was a survey used in study 4? Paper or online? What software was used?

Was a survey used in study 5? Paper or online? What software was used?

The conclusions need some work. When addressing the major findings, the followings are to be included: the contribution of the study results to the literature; the implications of the results for academicians and others; how the findings will be useful to the profession and audiences such as companies, consumers, and educators or service providers; and what additional research is needed in this area. Guidance to future researchers concerning method components is appropriate. Issues raised in the introduction and review of literature. Please consider these points when reworking.

Round 2

Reviewer 1 Report

Comments and Suggestions for Authors

Now the paper is much better.

Good luck!

Author Response

Thank you very much for your comments and feedback on this manuscript. We are pleased to hear that our revisions have been much better. Your support and encouragement motivate us to continue our research endeavors with dedication and diligence. Thanks again for your valuable comments and wishes. Your encouragement and support mean a lot to us.

Reviewer 2 Report

Comments and Suggestions for Authors

Thank you very much for taking the time to address each of my suggestions. I do believe the final discussion is still lacking in the sense of discussing the overall conclusions of each study. What is the "so what" of each?

Thank you again!

Author Response

Response 2: Thank you for providing valuable feedback on our manuscript. We may not have discussed the overall conclusions of each study in the final discussion section enough, which confused you. As suggested by you, we further revised the content in the final discussion. For more details of the changes, please see the red front on page 15, paragraph 4, lines 639-659.

“Study 1 used a haunted house as an experiential stimulus to evaluate participants' attitudes toward counterhedonic consumption. Compared to other counterhedonic consumptions (e.g., horror games and movies), this immersive experience of the haunted house is perceived to have a higher degree of authenticity[1]. In Study 2, we used a horror game as a stimulus, excluding the interference of authenticity. Haunted houses and horror games are usually counterhedonic consumptions in which multiple people participate together and have interactivity[2]. To mitigate potential interference of interactivity, Study 3 used horror movies as stimuli. The findings from Studies 1-3 indicated the effect of upward comparisons on counterhedonic consumption. Upward comparisons may influence all consumption behavior and not only counterhedonic consumption. Study 4 endeavored to address this question by examining the effect of upward comparisons on other types of consumption behaviors. Study 5 demonstrated the moderating effect of comparison targets. Specifically, when the comparison target was a friend, upward comparisons led to counterhedonic consumption. When the comparison target was a stranger, upward comparisons (vs. non-upward comparisons) do not significantly influence counterhedonic consumption.”

Thank you again for your suggestion.

  1. Lee, Y.J. Tourist behavioural intentions in ghost tourism: The case of Taiwan. INTERNATIONAL JOURNAL OF TOURISM RESEARCH 2021, 23, 958-970, doi:10.1002/jtr.2456.
  2. Habel, C.; Kooyman, B. Agency mechanics: gameplay design in survival horror video games. DIGITAL CREATIVITY 2014, 25, 1-14, doi:10.1080/14626268.2013.776971.